# On-Chip Synthesis of Hyaluronic Acid-Based Nanoparticles for Selective Inhibition of CD44+ Human Mesenchymal Stem Cell Proliferation

**DOI:** 10.3390/pharmaceutics12030260

**Published:** 2020-03-13

**Authors:** Enrica Chiesa, Federica Riva, Rossella Dorati, Antonietta Greco, Stefania Ricci, Silvia Pisani, Maddalena Patrini, Tiziana Modena, Bice Conti, Ida Genta

**Affiliations:** 1Department Drug Sciences, University of Pavia, V.le Taramelli 12, 27100 Pavia, Italy; enrica.chiesa@unipv.it (E.C.); rossella.dorati@unipv.it (R.D.); antonietta.greco01@universitadipavia.it (A.G.); tiziana.modena@unipv.it (T.M.); bice.conti@unipv.it (B.C.); 2Department of Public Health, Experimental and Forensic Medicine, Histology and Embryology Unit, University of Pavia, Via Forlanini 10, 27100 Pavia, Italy; federica.riva01@unipv.it (F.R.); stefania.ricci01@universitadipavia.it (S.R.); 3Polymerix srl, V.le Taramelli 24, 27100 Pavia, Italy; 4Immunology and Transplantation Laboratory, Pediatric Hematology Oncology Unit, Department of Maternal and Children’s Health, Fondazione IRCCS Policlinico S. Matteo, 27100 Pavia, Italy; silvia.pisani01@universitadipavia.it; 5Department of Physics, University of Pavia, Via Bassi 6, 27100 Pavia, Italy; maddalena.patrini@unipv.it

**Keywords:** hyaluronic acid-based nanocarriers, Everolimus, chitosan, microfluidics, CD44 targeting, human mesenchymal stem cells

## Abstract

In this study, an innovative microfluidics-based method was developed for one-step synthesis of hyaluronic acid (HA)-based nanoparticles (NPs), by exploiting polyelectrolytic interactions between HA and chitosan (CS), in order to improve reliability, reproducibility and possible scale-up of the NPs preparation. The on-chip synthesis, using a staggered herringbone micromixer, allowed to produce HA/CS NPs with tailored-made size and suitable for both parenteral (117.50 ± 4.51 nm) and loco-regional (349.15 ± 38.09 nm) administration, mainly composed by HA (more than 85% wt) with high negative surface charge (< −20 mV). HA/CS NPs were successfully loaded with a challenging water-insoluble molecule, Everolimus (EVE), an FDA- and EMA-approved anticancer drug able to lead to cell cycle arrest, reduced angiogenesis and promotion of apoptosis. HA/CS NPs resulted to be massively internalized in CD44+ human mesenchymal stem cells via CD44 receptor-mediated endocytosis. HA/CS NPs selectiveness towards CD44 was highlighted by blocking CD44 receptor by anti-CD44 primary antibody and by comparison to CS-based NPs cellular uptake. Eventually, high effectiveness in inhibiting cell proliferation was demonstrated on-chip synthetized EVE loaded HA/CS NPs by tracking in vitro DNA synthesis.

## 1. Introduction

Cluster of differentiation-44 (CD44) is a class of non-kinase, transmembrane glycoproteins widely distributed in normal adult and fetal tissues (i.e., brain, hematopoietic and connective tissue) and primary involved in multiple signaling pathways contributing to cellular differentiation, proliferation and migration along with metabolic shift, both in physiological and pathological processes [1,2]. CD44 and CD44-like receptors are also identified as molecular markers for human mesenchymal stem cells (hMSCs). They are involved in regulating stem cell recruitment during tissue development and regeneration as well as markers for cancer stem cells, and they are correlated with tumor progression, metastasis and chemo- and radio-resistance [3,4,5]. For these reasons tremendous research efforts have been focused on studying the diagnostic and prognostic value of CD44 in cancer and how to exploit CD44 as target agent in cancer therapy [6]. Up to now, the various strategies developed mainly involved specifically-designed antibody in order to antagonize CD44 pathway [7], or hyaluronic acid (HA)-based nanostructures for selective CD44 targeting. This last approach uses HA, that represents the main natural ligand of CD44, to develop CD44 selective antineoplastic drug-HA conjugates, HA coated liposomes/nanoparticles, self-assembled HA or HA-based nanoparticles (NPs) [7,8,9,10].

HA is a ubiquitous and highly hydrated polyanion belonging to the glycosaminoglycans family and composed of disaccharide units of D-glucuronic acid and *N*-acetyl-*D*-glucosamine. It is an important component of the extracellular matrix and its structural and biological properties mediate cellular signaling, wound repair, morphogenesis and matrix organization [11]. HA is widely used in the pharmaceutical field since it is a biocompatible, biodegradable and non-immunogenic negatively charged mucopolysaccharide that can be obtained both from animal source or bacterial bio-fermentation using modified organism. Many HA-based medical products for oral administration or local injection are already approved by European and US Regulatory Agencies and are on the market as viscosupplement agents for treatment of joint disorders, including osteoarthritis in humans, or well-established adjuncts to ophthalmic and plastic surgery [12].

The rationale of using HA-based NPs for targeted therapy towards CD44-expressing (CD44+) cells is threefold. First, HA ensures interaction with the specific HA-binding domain, located in N-terminal region of the extracellular domain of all isoforms of CD44, and thereby, HA can induce a CD44 targeted action and consequently trigger NPs cellular uptake leading to successful delivery of drugs into tumor cells [13,14]. Secondly, HA can provide a relative “stealth” feature to NPs preventing unspecific protein adsorption on NPs, and thereby allowing to opsonisation elusion and complement activation [15,16,17]. Lastly, thanks to the free carboxylic groups in the polymer backbone, HA-based NPs can be further chemically conjugated with an aptamer or other specific ligands in order to obtain dual/multi-targeted NPs which can increase selectivity and efficacy against CD44+ cancer cells [18,19].

HA-based NPs were traditionally obtained by ionotropic gelation methods based on polyelectrolytic interaction between oppositely charged polymers [9]. Chitosan (CS) is a natural polysaccharide obtained from renewable animal or vegetable sources, and it has been the most extensively used cationic polymer combined to HA in HA-based NPs (HA/CS NPs) preparation [20,21,22,23,24]. Thanks to its well-known non-toxicity, biocompatibility, biodegradability and low immunogenicity, CS recently gained FDA-approval for human use. CS suitability as nanomedicine ingredient is further broadened by its biological properties like mucoadhesion, antimicrobial, antifungal and antioxidant activities, but above all its inherent capacity to transiently open tight junctions [20,25]. HA/CS NPs formation is usually obtained by bulk mixing methods through dropwise addition of an HA solution, containing sodium tripolyphosphate (TPP), to a CS solution, under magnetic stirring. Agitation is maintained for few minutes, to allow the complete formation of HA/CS polyelectrolyte complex in form of spherical NPs (HA/CS NPs) [26,27]. This bulk mixing method allows to yield NPs with positive or slight negative surface charge indicating an incomplete HA coating; alternative inverse one-step bulk gelation procedure, bulk direct complexation and template-based method were recently proposed with a slight improvement of HA coating [28,29]. However, all the conventional bulk mixing methods are characterized by a lack of control over mixing, low batch-to-batch reproducibility and slow production rate. These issues usually result in the production of poorly defined NPs with heterogeneous composition and size, and unreliable in vivo performance that limit their translation towards clinical investigation [30].

To address the issues, microfluidics is emerging as promising diversified technology for NPs synthesis in a well-controlled, reproducible and high-throughput manner [31,32,33]. It improves the quality of NPs through an accurate control of the reaction environment and ensures a rapid screening of NPs in R&D phase by using small amounts of reagents. Thereby this kind of devices have the potential to decrease production costs and time, and to help NPs preparation process scale-up so bridging the gap between bench to bedside [33,34].

At the best of our knowledge, no papers have been published dealing with on-chip synthesis of HA-based NPs by ionotropic gelation; first attempt on this topic was flow focused nanoprecipitation technique by a non-solvent extraction coupled with HA crosslinking reaction based on divinylsulfone or with self-assembling of the peptide novocidin with an octenyl succinic anhydride-modified HA analogue [35,36,37,38].

Our work aimed to develop a microfluidics-based method for one-step synthesis of tailor-made HA/CS NPs with improved quality and reproducibility. The optimization was aimed towards obtaining HA-based NPs with customized features and uniform HA coating in order to achieve potentially strong active targeting towards CD44+ cells.

HA/CS NPs were loaded with a challenging water-insoluble drug, Everolimus (EVE), an antiproliferative agent currently approved by many agencies for treatment of advanced cancers [39], most of them characterized by CD44-overexpressing feature. The suitability of on-chip microfluidic approach for HA/CS NPs synthesis was demonstrated by high EVE loadings and its associated release behavior, and it was corroborated by in vitro studies on the NPs ability to effectively cross the hMSC membrane, via CD44 receptor, releasing EVE in the cell cytosol and notably reducing the proliferative activity of CD44+ cells.

## 2. Materials and Methods

### 2.1. Materials

Hyaluronic acid sodium salt (HA, high molecular weight: 750 kDa) and sodium tripolyphosphate (TPP) were purchased from Sigma Aldrich (St. Louis, MO, USA). Chitosan chloride salt (CS) pharmaceutical grade, (Chitoceuticals, viscosity 19 mPa (1% in water), deacetylation degree 82.2%, chloride content 13%) was supplied from Heppe Medical Chitosan GmbH (Halle, Germany). Everolimus (EVE, Mw 958.22 g/mol) was obtained from AbMole Bioscience (Houston, TX, USA).

Anti-mouse IgG secondary antibody Fluorescein IsoThioCyanate (FITC) conjugated, Thiazolyl Blue Tetrazolium Bromide (MTT, approx. 98% TLC), 5-Bromo-2′-deoxyuridine (BrdU), Dulbecco’s Modified Eagle’s Medium High glucose (DMEM), Dulbecco’s Phosphate Buffered Saline (PBS 10X, sterile), Cibacron Brilliant Red 3B-A (dye content 50%), Hoechst 33258 solution and Penicillin-Streptomycin Solution (100×), were obtained from Sigma Aldrich (St. Louis, MO, USA). Anti-CD44 (Phagocytic glycoprotein-1, HCAM) monoclonal mouse primary antibody, anti-human CD44 and anti-mouse IgG secondary antibody FITC conjugated were purchased from Biogenex (San Ramon, CA, USA). Fetal bovine serum was from EuroClone Spa (Milan, Italy). Monoclonal Anti-BrdU antibody produced in mouse was obtained from Amersham-GE-HealthCare UK Limited (Buckinghamshire, UK).

Unless otherwise noted, water was distilled and filtered (0.22 μm membrane filters, Millipore Corporation, Billerica, MA, USA) and chemicals were of analytical or HPLC grade.

### 2.2. Cell Lines

hMSCs from human bone marrow were kindly provided by Fondazione IRCCS Policlinico San Matteo (Paediatric Oncohaematology Unit, Pavia, Italy).

### 2.3. Microfluidic Device

A staggered herringbone micromixer (SHM) was employed connected to the automated mixing platform (NanoAssemblr™, Precision NanoSystems Inc., Vancouver, BC, Canada). Polypropylene, viton and cyclic olefin copolymer-based microfluidic cartridges are characterized by a Y-shaped architecture with mixing channels of 200 × 79 μm (wxh) incorporating staggered herringbone ridges (31 μm high and 50 μm thick) lithographed on the floor of the channel with an angle of 45° with respect to the long axis of the channel [40]. The characteristic mixing cycle consist of two sequential regions of ridges with herringbone structure; the asymmetric ridges (respect to the channel centerline) change their direction from one region to the next. The ridges are liable of a chaotic flow and another important feature is the reduction of the mixing length due to the decrease of the average distance over which diffusion acts in the transverse direction to homogenize unmixed volumes [41]. Disposable and compatible syringes were used to inject solutions into separate inlet points on the Y-shape micromixer. The process parameters selected were Total Flow Rate (TFR), intended as sum of the flow rates of two inlet solutions, and relative Flow Rate Ratio (FRR), i.e., volume ratio between the two inlet solutions; sample batches were collected from the outlet point.

### 2.4. HA/CS NPs on-Chip Preparation Method Set-up

CS aqueous solution (1.5 mL) and a mixture made of HA and TPP aqueous solutions (1.5 mL) were injected into separate chambers using 5 mL BD syringes (Becton Dickinson Italia S.p.A., Milano, Italia) housed in the appropriated compartments. Then, 3 mL of sample batches were recovered from the outlet port; initial and ended waste were set at 0.350 mL and 0.050 mL, respectively, with a 2.6 mL core sample. TFR was kept constant at 12 mL/min as well as the volume ratio between CS and HA/TPP solutions was unchanged (FRR 1:1).

In the suitable experimental design space (Appendix A), a deeper investigation was performed to critically evaluate the role of raw materials in producing HA/CS NPs with tunable size distribution (150–500 nm mean size, a polydispersity index (PDI) < 0.400) and high negative surface charge, that was taken as indicator of HA coating effectiveness.

HA concentration and CS:TPP weight ratio were chosen as the critical formulation variables and they were systematically varied: HA aqueous solution concentration from 0.4 to 0.15 mg/mL and CS:TPP weight ratios of 50:1, 25:1 and 12.5:1 *w*/*w*.

All HA/CS NPs batches were recovered after centrifugation at 16,400 rpm at 4 °C for 30 min (Eppendorf Centrifuge 5417 R, Eppendorf s.r.l., Milan, Italy).

### 2.5. Optimization of HA/CS NPs Preparation Method

SHM-assisted ionotropic gelation method previously set-up was duly amended to load EVE, a poorly water-soluble drug [38]. Considering both EVE solubility characteristics and solvent classification as reported in ICH Q3C “Impurities: guidelines for residual solvents”, ethanol (Et) and methanol (Me) were selected as the suitable organic solvents since they belong to class 3 and 2, respectively.

Briefly, an aliquot (5% *v*/*v*) of organic solvent was added to the CS aqueous solution in order to simulate drug solution and placebo HA/CS NPs were prepared by separately pushing hydroalcoholic CS solution and aqueous HA/TPP solution through the two microchannels of the SHM setting. Process conditions were applied as set-up in Section 2.4.

### 2.6. Everolimus Loaded HA/CS NPs

EVE ethanolic solution (0.2 and 0.5 mg/mL) was added to CS aqueous solution (5% *v*/*v*); two different CS:drug weight ratio were tested, namely 5:1 and 2:1 *w*/*w*.

Hydroalcoholic CS/EVE solution and aqueous HA/TPP solution were separately pushed through the two microchannels of the SHM setting following the set-up process conditions as in Section 2.5.

### 2.7. Characterization of NPs

The mean hydrodynamic diameter, PDI and zeta (ζ) potential of blank, of placebo and EVE loaded HA/CS NPs were measured by dynamic light scattering (DLS) using NICOMP 380 ZLS (Particles Sizing System, Santa Barbara, CA, USA). Analyses were performed in triplicate for each composition tested and results reported as mean ± SD.

Morphometric analysis of NPs was performed by transmission electron microscopy (TEM, EM 2085, Philips EindHoren, Holland); the collected images were processed by Jmicrovision software to evaluate NPs sizes.

HA and CS contents into NPs were determined following methods found in the literature [28]. Briefly, an HPLC method based on the size exclusion liquid chromatography with UV detection was used to detect the amount of HA in the supernatant, left after NPs centrifugation. The chromatographic experiments were performed on Agilent 1260 infinity HPLC with Yarra SEC 2000 column (300 mm × 7.8 mm column, with 3 μm silica particle size and 145 Å pore size) at 25 °C. Potassium dihydrogen phosphate aqueous solution (pH 7) was used as isocratic mobile phase at a flow rate of 1 mL/min and the detection wavelength was set up at 205 nm.

HA association efficiency (% HA) was calculated as:% HA = 100 × (HA_(tot)_ − HA_(sup)_)/HA_(tot)_(1)
where HA_(tot)_ is the starting amount of HA, while HA_(sup)_ is the amount of HA detected in the supernatant after NPs suspension centrifugation.

CS content was determined by Cibacron Brilliant Red 3B-A colorimetric assay [28]. The supernatants (100 µL) were collected by centrifugation and diluted with 200 μL of glycine buffer and 3 mL of a dye solution (1.5 g/L). A final sample was analyzed using a UV–VIS spectrophotometer (6705 UV–VIS spectrophotometer, Jenway, Staffordshire, UK) at 575 nm. A solution made of glycine buffer (300 µL) and dye solution (3 mL) was used as reference.

CS association efficiency (% CS) was calculated as:% CS = 100 × (CS_(tot)_ − CS_(sup)_) / CS_(tot)_(2)
where CS_(tot)_ is the starting amount of CS, while CS_(sup)_ is the amount of CS detected in the supernatant.

FTIR spectra on placebo NPs, raw polymers (CS, HA) and TPP were carried out in order to characterize the interactions between HA and CS biopolymers in HA/CS NPs. The ATR mode with a Ge crystal was used. FTIR spectra were recorded using a Thermo Scientific Nicolet iN10 spectrometer (Waltham, MA, USA) at 256 scans and a resolution of 1 cm^−1^. Before analysis, all NPs were freeze-dried at −50 °C, 0.01 bar for 24 h (Lio 5P, Cinquepascal s.r.l., Milano, Italy).

EVE content in the NPs was measured by HPLC analysis of the supernatant after EVE loaded HA/CS NPs centrifugation [42]. HPLC (Agilent 1260 Infinity, Agilent, Santa Clara, CA, USA) equipped with a Zorbax Eclipse^®^ Plus C18, 4.6 × 150 mm, 5 μm coupled with a precolumn was used and heated at 55 °C. Detection wavelength was 278 nm. Mobile phase made of water (22% *v*/*v*), acetonitrile (18% *v*/*v*) and methanol (60% *v*/*v*) was eluted at the flow rate of 1.5 mL/min. EVE content was determined against EVE standard solution at concentration range 1–20 μg/mL (*R*^2^ = 0.9997). Results were averaged over six individual experiments and reported as drug loading (DL), expressed as EVE amount in 1 mg of NPs, and as encapsulation efficiency percentage (EE%) ± SD.

In vitro release study was performed in *sink* conditions as follows: 100 μg of EVE loaded HA/CS NPs (containing about 5.8 and 8.8 µg total EVE content) were resuspended in 720 µL of 0.01M PBS with 1% *w*/*v* Tween 20 (pH 7.4) and incubated at constant temperature of 37 °C under gentle shaking. At scheduled time points, NPs samples were centrifuged (16,400 rpm, 30 min, 4 °C), supernatant was withdrawn and analyzed by HPLC (as reported before). Results were expressed as mean of EVE released percentage ± SD (*n* = 3). EVE as such underwent a dissolution test in the same experimental conditions.

### 2.8. Immunocytochemistry Assay

CD44 expression on hMSCs surface was labeled by an immunocytochemistry (ICC) assay as previously described [43].

Briefly, hMSCs, cultured on bottom glass slides, were incubated for 20 min at room temperature with 500 μL of PTA blocking solution made of 0.01 M PBS supplemented with Tween 20 (0.02% *w*/*v*) and Albumin (1% *w*/*v*). An aliquot of antibody solution (anti-CD44/PTA, 40 μL) containing 1.5 µg/mL of mouse primary antibody anti-CD44 was dropped on each bottom glass slides and they were incubated at room temperature, for 1 hour. Afterwards, the samples were washed twice with PTA blocking solution and incubated for 30 min in the darkness, at room temperature with 1 μg/L FITC-labeled anti-mouse secondary antibody in PTA solution (drop of 40 μL for each sample). Subsequently, all samples were washed twice with PTA and stained by 0.5 µg/mL Hoechst33258, a specific dye for nuclear DNA. After a final washing with PBS, the samples were observed at confocal laser scanning microscopy (Leica TCS SP8 AOBS^®^, Wetzlar, Germany). Various images were captured in different optical field of each sample. All images were taken at 40× objective magnification.

### 2.9. Cellular uptake of HA/CS NPs

hMSCs were seeded on bottom glass slides (20,000 cells/well) and cultured in DMEM added with FBS and antibiotics (at 37 °C with 5% CO_2_) till reaching 60–70% of cell confluence. Then, cells were treated with 500 μL of placebo Rhodamine (RhB)-conjugated HA/CS (HA/CS–RhB) NPs suspension, prepared as described in Section 2.9.2, at different concentrations (50 µg and 100 µg/20,000 cells). RhB-conjugated CS (CS-RhB/TPP) NPs (see Section 2.9.3) were used as control whereas untreated cells were used as negative control.

At scheduled time points (30, 90 and 240 min) all media were removed and the cells were washed with sterile PBS and fixed with 4% *w*/*v* paraformaldehyde aqueous solution. Eventually, paraformaldehyde was removed, the cells were washed twice with PBS and nuclei were stained with Hoechst. The cells were viewed under confocal microcopy (Leica TCSSP8, AOBS, Germany, obj. mag. 40×) and all confocal images were elaborated by ImageJ software to determine the level of fluorescence due to the presence of NPs inside the cells obtaining a quantitative evaluation of the uptake of fluorescent NPs into hMSCs. Quantitative analysis was carried out analyzing at least 30 cells for each sample and results were expressed as Corrected Total Cell Fluorescence (CTCF) calculated by Equation (3):CTCF = Integrated density − (Area of selected cell × Mean fluorescence of background readings)(3)

Integrated density is the product of Area and Mean gray: the value is provided by the Set Measurement of software ImageJ Version 1.52a.

#### 2.9.1. Chitosan-Rhodamine B Conjugation

Exploiting the high density of positive charges primary amino groups on CS backbone, RhB was grafted to CS through an amidation reaction using 1-Ethyl-3-[3-(dimethylamino) propyl] carbodiimide hydrochloride (EDC) and N-hydroxysuccinimide (NHS) as reagent to form CS-RhB conjugate with strong red fluorescence [40]. CS-RhB conjugation reaction was performed as follows: 490 mg of RhB (0.001 mol), 390 mg of EDC (0.002 mol) and 230 mg of NHS (0.002 mol) were dissolved in 20 mL of PBS pH 5.8. Afterwards, 20 mL of CS solution 1% (*w*/*v*) dropwise were added; pH was adjusted to 5.0 with 1M NaOH aqueous solution and stirred in the dark at room temperature for 24 h. The reacted mixture was purified by dialyzing (MWCO: 12–14,000 Da, SPECTRA/POR^®^) against demineralized water for 3 days in the dark, with twice a day water changes. Finally, the purified product was freeze-dried.

The reaction yield after purification was gravimetrically determined by the following expression:Y (%) = 100 × CS − RhB/(CS + RhB)(4)
where CS-RhB is the amount (mg) of the fluorescently labeled conjugate, RhB and CS are the starting raw materials amount (mg). CS-RhB conjugation was confirmed by UV–VIS spectrophotometer (Jenway™ 6705 Model), UV-vis spectrum of CS-RhB was scanned between 200 nm and 700 nm, pristine CS and RhB were used as control [44]. RhB labeling efficiency was determined by measuring the absorbance of CS-RhB conjugate solution with respect to RhB standard solutions at 554 nm in the concentration rank 0.25–4 µg/mL (*R*^2^ = 0.9991).

#### 2.9.2. Placebo HA/CS-RhB NPs Preparation

A physical blend made of CS (66% wt) and CS–RhB conjugate (34% wt) was dissolved in water to reach the final concentration of 0.05 mg/m whereas HA and TPP were separately dissolved in water and mixed together to reach the final concentration of 0.15 mg/mL and 2 µg/mL, respectively. HA/CS–RhB NPs were prepared through NanoAssemblr^TM^ platform following the protocol described in Section 2.5.

#### 2.9.3. Placebo CS-RhB/TPP NPs Preparation

CS (98% wt) and CS–RhB conjugate (2% wt) were dissolved in water to reach the final concentration of 2 mg/mL. TPP was dissolved in water at 0.5 mg/mL concentration. CS-RhB/TPP NPs were prepared through NanoAssemblrTM platform: 5 mL of CS-RhB aqueous solution (2 mg/mL) and 5 mL of TPP aqueous solution (0.5 mg/mL) were injected into the microfluidics device. An aliquot of 3 mL of sample were recovered from the outlet; initial and ended waste were set at 0.350 mL and 0.050 mL, respectively, indeed the core sample was about 2.6 mL. TFR and FRR were kept constant at 8.5 mL/min and 1:1 (*v*:*v*), respectively.

### 2.10. Competitive Binding Experiment

A competitive binding experiment was performed on the hMSCs. Briefly, hMSCs were pre-treated with 1.5 µg/mL of mouse anti-CD44 glycoprotein primary antibody solution for 30 min at 4 °C and then cells were incubated at 37 °C, 5% CO_2_ [45]. After 24h, the cells were treated with HA/CS-RhB NPs and CS-RhB/TPP NPs (50 µg/20,000 cells) for 240 min at 37 °C following double washing with sterile PBS, fixation with 4% *w*/*v* paraformaldehyde aqueous solution and 10 min incubation.

Cell nuclei were stained with Hoechst (0.5 µg/mL), ICC assay was then performed to highlight CD44 receptors on the cell membrane.

### 2.11. In Vitro Cytotoxicity

An IC50 study was carried out to determine the toxicity of EVE loaded into HA/CS NPs towards CD44+ cells and with respect to free EVE. hMSCs were used and seeded in 96-well plate, at the density of 10,000 cells per well, for 24 h. Then the medium was discharged, the cells were washed with PBS and incubated with EVE loaded HA/CS NPs (0.1–1000 µg/mL in culture medium) or free EVE at equivalent drug concentration (0.01 to 88.12 µg/mL in culture medium). Placebo HA/CS NPs in concentration rank between 0.1 to 1000 µg/mL were used as control. Samples were incubated with cells at 37 °C in a humidified atmosphere containing 5% CO_2_ for 24 h.

In vitro cell viability was determined by using Trypan Blue dye method. Briefly, hMSCs were detached using trypsin (25% trypsin/2.21 mM EDTA) and the viable cells were highlighted by Trypan Blue dye and counted by using Burker chamber.

The collected cells number were used to create percent viability vs. dose curves with the PRISM 6.0 program from GraphPad Software Inc. (La Jolla, CA, USA) using the sigmoidal dose-response curve fit. IC50 values were then determined from the fitted curves.

### 2.12. Assessment of DNA Synthesis and Proliferative Activity by 5-Bromo-2′-Deoxyuridine Incorporation

Antiproliferative effect of EVE loaded HA/CS-TPP NPs on hMSCs was evaluated analyzing DNA synthesis by measuring the incorporation of 5-bromo-2′-deoxyuridine (BrdU) [46].

About 20,000 cells (500 μL seeding volume) were seeded on round glass slides (∅ 12mm) and cultured in growth DMEM medium with 10% of FBS and 1% antibiotics up to reach 60–70% of confluence. Cells and EVE loaded HA/CS NPs were incubated at 37 °C (with 5% CO_2_) for 30, 90 and 240 min. Different EVE loaded HA/CS NPs concentrations were tested, namely 5 μg, 10 μg, 25 μg and 50 μg/20,000 hMSCs. Placebo HA/CS NPs (10 μg and 50 µg/20,000 hMSCs) and free EVE (2.2 µg/20,000 cells) were used as positive control. Untreated cells (CTR) were the negative control.

About 1 h before the end of the test, the medium was discarded and cells were treated with an aliquot (500 μL) of 30 µM BrdU diluted with fresh DMEM for 45 min incubation at 37 °C, 5% CO_2_. After incubation, fixing procedure was performed: medium was removed and cells, washed twice with sterile PBS, were fixed using 4% *w*/*v* paraformaldehyde aqueous solution, washed twice with PBS and stored at −20 °C.

BrdU immunostaining was performed as described in Riva et al. [43]. Briefly, coverslips were washed with PBS, incubated with 500 μL of 2N chloric acid for 30 min at room temperature to hydrolyse DNA and then washed with 500 μL of 0.1 M sodium tetraborate neutralizing solution (pH 8.5) for 15 min. After two washes with PBS for 5 min, the samples were incubated for 20 min with a PTA blocking solution (1% *w*/*v* BSA and 0.02% *w*/*v* Tween 20 in PBS) and, subsequently, for 1 h with mouse anti-BrdU antibody, diluted 1:100 *w*/*v* in PTA solution. Samples were washed 3 times (10 min each) with the same solution, and then incubated in the dark for 30 min in PTA solution containing anti-mouse IgG FITC-conjugated antibody (Sigma-Aldrich, 1:100 *v*/*v* dilution). Finally, all samples were extensively washed with PTA and quickly with PBS and nuclei were counterstained for DNA with 0.5 μg/mL Hoechst 33258 (Sigma-Aldrich). Samples were observed by means a Zeiss Axiophot fluorescence microscope (Carl Zeiss, Oberkochen, Germany; blue filter: λ_ex._ = 346 nm and λ_em_ = 460 nm; green filter: λ = 494 nm and λ_em_ = 518 nm). A total of at least 500 cells with blue stained nuclei was counted for each condition. Replicating cells were scored for BrdU incorporation detectable as green immunofluorescence of positive nuclei. Proliferative activity was expressed as a percentage obtained by the number of proliferating cells (BrdU positive nuclei) in relation to the total number of counted viable cells.

### 2.13. Statistical Analysis

Unless otherwise indicated, the experiments were repeated at least three times. Results were given as mean value ± SD, multiple *t*-test or one-way (ANOVA) with either Sidak’s or Tukey’s multiple comparison analysis were performed by GraphPad Prism software (Graphpad software, Version Prism 6). *p* value < 0.05 was considered statistically significant.

## 3. Results and Discussion

Polyelectrolytic interactions between negatively charged groups of a polyanion, such as HA, and the positively charged primary amino groups of CS, fostered by a small ion with triple negative charge such as TPP, underlie the ionotropic gelation technique, an eco-friendly and safe NPs preparation method [47]. The same reactions were exploited for the set-up of on-chip HA/CS NPs synthesis through a microfluidic approach using an SHM device. By inducing a chaotic advection mixing profile of the inlet fluids, via Y-junction, the SHM pattern causes repeated rapid folding of the two solution streams increasing the interface surface area and the rate of mixing, and concomitantly reducing the diffusional distance [41].

On-chip synthesis set-up of HA/CS NPs was developed through a precise and reproducible manipulation of extremely small amounts of reagents and an accurate control of the reaction environment. The process was studied to enable NPs manufacture with required physicochemical properties, namely tunable sizes with narrow size distribution and high HA deposition on NPs surface. From a technological stand point, if it is well-known the influence of NPs size and surface properties in their interaction with biological systems, it is crucial to develop versatile methods that accurately control these parameters in order to enable reproducible high scale production for clinical investigation.

### 3.1. HA/CS NPs On-Chip Preparation Method Set Up

On-chip synthesis of HA/CS NPs through ionotropic gelation method involves an extremely fast mixing, in the millisecond range, of two miscible solution (HA/TPP and CS aqueous solutions).

First, set-up of a NPs preparative procedure goes through to establish the best ratio between the compounds in order to obtain a suitable final product. Thus, preliminary experiments were carried out to broadly identify the suitable conditions for NPs formation (Appendix A, Figure A1). As well-known, the HA/CS NPs formation mechanism is due to polyelectrolytes complexation between both polysaccharides, CS and HA with opposite charge, and CS ability to undergo a liquid-gel transition in presence of TPP [22,23,24,26,28,29].

NPs size and negative ζ potential, meaning that HA effectively covered the NPs surface, were considered as output at first glance.

The selected HA concentrations were 0.400 mg/mL and 0.150 mg/mL, instead, CS and TPP concentrations were set at 0.100 and 0.050 mg/mL and 4 or 2 μg/mL, respectively, and thus the screened CS/TPP weight ratios were 50, 25 and 12.5:1 *w*/*w*.

HA/CS NPs were characterized in terms of mean size, PDI and ζ potential, the data are shown in Table 1.

Statistically analyzed results (Sidak’s multiple comparisons test, *p* value < 0.05) revealed the noteworthy role of HA concentration and HA:CS weight ratio on the HA/CS NPs features.

At first instance, HA concentration affects HA/CS NPs sizes; in particular, its increase (0.400 mg/mL) triggers an important NPs enlargement (580–1000 nm). Moreover, HA:CS weight ratio is responsible of the NPs’ negative surface charge. HA:CS weight ratio higher than 1.5:1 (BL_1, BL_4 - BL_8, Table 1) is liable for NPs ζ potential values around -20mV, suggesting a uniform HA deposition on NPs surface that is crucial for CD44 targeting.

More in details, positive surface charge of +9.30 ± 10.89 mV and +17.34 ± 4.13 mV was observed for BL_2and BL_3 prepared by using 0.150 mg/mL HA solution combined to 0.1 mg/mL CS solution (HA:CS ratio 1.5:1 *w*:*w*) and CS:TPP ratios of 50:1 and 25:1 *w*/*w*. These charge values highlight a predominant CS exposure on the NPs surface, that is an unfavorable condition to perform CD44 target. Nevertheless BL_2 and BL_3 showed suitable particles size of 102.75 ± 3.23 nm and 159.85 ± 38.91 nm with PDIs of 0.297 ± 0.007 and 0.318 ± 0.039. These results show a narrow size distribution for all batches with only a slight decrease in particles size by using the highest CS:TPP ratio (50:1 *w*/*w*).

In order to obtain an efficient HA coating on NPs an higher HA:CS ratio (3:1) was tested: the lowest HA and CS concentrations were combined together with two different CS:TPP ratio, namely 12.5:1 and 25:1 *w*:*w* (BL_6 and BL_7, Table 1). The NPs samples were characterized by a high negative potential of about −20 mV; moreover, suitable NPs mean size of 239.20 ± 19.51 nm and 159.00 ± 7.28 nm were obtained for BL_6 and BL_7, respectively, and statistically significant reduction in particle size was observed for 25:1 CS:TPP ratio (BL_7, *p* value = 0.0026 by using Holm-Sidak method). A satisfactory PDI value of 0.306 ± 0.053 was recorded for BL_7 while a not completely satisfactory one for BL_6 (0.520 ± 0.059).

With the aim to maximize HA exposure on the NPs surface, HA concentration was increased up to 0.4 mg/mL by keeping the lowest CS concentration (HA:CS weight ratio 8:1) and the same CS:TPP ratios (12.5:1, 25:1 *w*:*w*) (BL_4 and BL_5, Table 1). The same satisfactory negative surface charge (about −20mV) was detected for both samples. However, NPs revealed the largest sizes, namely around 900 nm when 25:1 *w*/*w* CS:TPP weight ratio was used (BL_5), that significantly decreased to 584.00 ± 22.20 nm when CS:TPP of 12.5 *w*/*w* was used (BL_4). PDI values around 0.400 were detected for the two batches.

A strong negative ζ potential of −24.88 ± 3.07 mV was also showed for BL-1 prepared using the highest HA and CS concentrations, HA:CS ratio 4:1 and CS:TPP ratio 25:1, indicating the consistent HA coating of NPs, even if NPs size was not acceptable (1025.45 ± 255.10 nm).

From the data collected for on-chip HA/CS NPs synthesis, at constant CS concentration (BL_4-7, Table 1) the lowest TPP concentration (CS:TPP 25:1 *w*:*w*) resulted not always responsible for NPs size enlargement. NPs’ formation was triggered by the randomized electrostatic interaction between HA and CS, and dimensional features are more likely correlated to HA concentration. Likewise, in bulk ionotropic gelation technique, TPP addition was needed to achieve more organized and more compact nanostructures, due to the strong electrostatic interactions between CS and TPP in the nanometer range, thus indicating that the CS/TPP ionic gelation is still required for the suitable NPs’ formation. In the literature, high CS:TPP weight ratios (15–20:1) were correlated to a slower kinetic of NPs formation [15,48]. This may mean a more controlled NPs formation process and, at the same time, the ability to promote CS/HA chains’ entanglement and ionic interactions to stabilize the nanostructures.

Our results and speculations were confirmed by BL_8 batch (Table 1) prepared using HA and CS solutions at intermediate concentration (0.275 and 0.075 mg/mL, respectively), corresponding to a 3.6:1 HA:CS weight ratio and maintaining the CS:TPP ratio at 25:1 *w*/*w*: NPs showed a mean diameter of 758.45 ± 23.77 nm with a PDI value of about 0.4 and they were characterized by a strong negative ζ potential (−24 mV).

BL_4 and BL_7 (Table 1), which would differ as much as possible in size but far below 1000 nm, were selected for further characterization, also considering they could be the most promising candidates for different administration routes such as parenteral vs loco-regional application.

Results of TEM analyses (Figure 1A, BL_4 and BL_7 NPs) showed spherical NPs with smooth surface, where BL_4 visual inspection showed less compact nanostructures. TEM micrographs were elaborated by Jmicrovision v 1.27 and mean size of 520.73 ± 99.59 nm and 199.22 ± 44.94 were recorded for BL_4 and BL_7, respectively. Therefore, consistent diameters were obtained by both DLS and TEM characterization.

Reacted HA and CS amounts were duly quantified in order to assess the NPs polymeric composition. For, CS amount involved in BL_4 formation resulted to be 19.66 ± 4.30 μg, corresponding to the 26.21% of CS starting raw material, while HA amount was 298.14 ± 11.69 μg, corresponding to 49.69% of HA starting raw material. BL_7 was composed by 28.75 ± 3.96 μg of CS, which is the 38.33% of CS starting raw material, and 219.36 ± 25.10 μg of HA corresponding to 97% of HA starting raw material. The results show that both BL_4 and BL_7 samples are mainly composed of HA, about 93% and 88%, respectively; the theoretically calculated negative/positive charges ratio (−/+) was 5.2 and 4, respectively, these data support the high negative ζ potential detected (about −20 mV) due to the existence of plenty of HA carboxylic groups on the NPs surface. This feature is primarily useful for the CD44 targetability, but also contributes to reduce plasma proteins absorption, avoid the quick capture by reticuloendothelial system and prolong the systemic circulation time.

In the second step, the set-up on-chip ionotropic gelation method was properly amended according to loading of EVE, a poorly water-soluble drug, into HA/CS NPs. Considering EVE chemical structure and solubility, ethanol (Et) and methanol (Me) were chosen as suitable organic solvents for EVE solubilization and alcoholic solution (5% *v*/*v*) was added to CS aqueous solution (see Section 2.5). Table 2 lists the results concerning placebo NPs prepared using Et (Et_CS_4 - Et_CS_7) or Me (Me_CS_4 - Me_CS_7): negligible differences were detected for placebo NPs batches prepared using Et or Me in terms of size distribution. Mean size of 349.15 ± 38.09 nm and 406.80 ± 19.65 nm were revealed for Et_CS_4 and Me_CS_4, respectively (Table 2). Same trend was observed for Et_CS_7 and Me_CS_7: Et addition led to the formation of NPs of 117.50 ± 4.51 nm (Et_CS_7, Table 2) that are fully dimensionally superimposable with those obtained by Me addition (Me_CS_7, mean size of 120.03 ± 13.92 nm, Table 2).

Recorded data suggest that organic solvent caused a slight decrease in NPs size if compared to the corresponding BL_4 and BL_7 NPs samples (Table 1), prepared by the same process conditions but without organic solvent.

All batches prepared maintained a strong negative ζ potential ranging from −20 to −31 mV.

Furthermore, impact of organic solvent addition on NPs polymer composition was verified and HA and CS amounts (μg) that successfully reacted to form placebo NPs are listed in Table 2. In all cases, Et and Me addition did not substantially modify NPs composition in terms of HA and CS amounts, also with respect to the corresponding BL_4 and BL_7: HA amount in all NPs batches was around 88 and 96%. Me_CS_4 revealed the most variable composition indicated by the high SD. A similar composition was revealed for Et_CS_7 and Me_CS_7: around 220 μg (93% of HA total amount) and 16 μg (about 7% of CS total amount), respectively. The highest HA content (about 96% wt) and the lowest CS content (about 4% wt) was detected in Et_CS_4.

Based on these data, Et was selected as solvent of choice for EVE; furthermore, Et belongs to class 2, as reported in ICH Q3C “Impurities: guidelines for residual solvents”, hence it has a lower toxicity than Me.

HA and CS interaction during NPs formation was further evaluated by comparing HA/CS NPs and raw materials FTIR spectra (Figure 1B,C).

The spectrum of the raw HA showed characteristic amide bands of the sodium form at about 1606, 1560, 1406 and 1322 cm^−1^ related to C=O bond, amide II, C–O bond of –COONa group and amide III, respectively [21,49]. The absorption wavenumbers of 1626 and 1520 cm^−1^ in the CS hydrochloride spectrum—typical of post-deacetylated CS—can be attributed to the C=O stretching vibration of amide I and to the N–H stretching vibration of amino group, respectively [50]. Bands at 1152 and 1028 cm^−1^ corresponds to symmetric stretching C–O–C and C–O stretching. TPP spectrum clearly shows characteristic bands at 1095 cm^−1^ (symmetric and symmetric stretching vibration of the PO_3_ groups), 1137 cm^−1^ (symmetrical and asymmetric stretching vibration of the PO_2_ groups) and 1210 cm^−1^ (P–O stretching) [51]. Several characteristic HA and CS vibrations are intensified and slightly shifted to higher wavenumbers in the spectrum of HA/CS NPs. The signal displacement confirms the mixture of both type of macromolecular chains in NPs formation; this is particularly evident in the structured band between 1520 and 1670 cm^−1^. A new shoulder arises at 1738 cm^−1^, associated with HA protonation occurring during the polyelectrolyte complex formation, as previously identified [21,52].

### 3.2. Everolimus Loading in HA/CS NPs

EVE loaded HA/CS NPs batches were prepared testing two different CS:EVE weight ratios: *i*) EVE-1_4 and EVE-1_7 with a CS:drug ratio of 2:1 *w*/*w* and, *ii*) EVE-2_4 and EVE-2_7 with CS:drug ratio was 5:1 *w*/*w*. The values of particle size, size distribution and ζ potential, reported in Table 3, are comparable to those of placebo HA/CS NPs, Et_CS_4 and Et_CS_7, respectively (Table 2).

EVE-1_4 disclosed mean size of 372.35 ± 71.50 with PDI mean value of 0.440 and a surface charge below to −24 mV, equivalent results were revealed for EVE-2_4 (mean size = 415.58 ± 84.56 nm, PDI = 0.490 ± 0.061, ζ potential = −24.61 ± 7.26 mV).

EVE-1_7 showed mean size of 144.93 ± 13.31 nm with PDI lower than 0.3 and a surface charge of −28.91 ± 6.18 mV while EVE-2_7 were characterized by a mean diameter of 135.07 ± 45.81, PDI value about 0.350 and surface charge −17.28 ± 2.30 mV.

EVE encapsulation did not significantly modify the NPs size characteristics. Consistent negative surface charge suggested that EVE was placed in the HA/CS NPs core.

Considering the challenging features of the drug and the hydrophilic polymeric network, satisfactory EVE loading was achieved for all batches: drug loading was increased from 30- to 90-fold if compared to EVE loading obtained in HA/CS NPs prepared by a traditional bulk mixing method [28].

DLs of NPs with the same HA and CS composition, namely EVE-1_4 - EVE-2_4 and EVE-1_7-EVE-2_7 (Table 3), were not affected by the two CS/EVE ratios tested, however the results demonstrate the relevant role of CS amount to obtain the highest EVE loadings. In particular, for the same CS/EVE ratio (EVE-1_7 vs. EVE-1_4 and EVE-2_7 vs. EVE-2_4) the NPs characterized by a greater CS content (EVE-1_7 and EVE-2_7) reached the highest DLs: in the presence of a non-ionic small drug this behavior can be ascribed to the role of CS molecules in the polyelectrolyte complexation of HA inside the microfluidic channel during the nucleation process typically depending on both anionic/cationic macromolecules concentration and size as well as their balance [35,53,54].

Nevertheless, for NPs with different polymer composition, a lower CS/EVE (2:1) ratio (EVE-1_4, EVE-1_7) induced a drop in the EE%, as previously reported in literature [55].

EVE-2_4 and EVE-2_7 reached the most satisfactory EE% (> 55%). These batches were thus considered for further investigation. Morphometric analysis of EVE-2_4 and EVE-2_7 (Figure 2A,B, respectively) shows NPs with spherical homogeneous shape while EVE-2_4 showed a less dense structure; JMicrovision software elaboration confirmed the DLS particle size data.

EVE in vitro release studies from EVE-2_4 and EVE-2_7 were carried out in 0.01M PBS with 1% *w*/*v* Tween 20 (pH 7.4). As known from the literature, the drug release from polysaccharides-based NPs is ruled by the NPs swelling phenomena combined to polymers electrostatic interactions breakdown due to medium ionic strength [28,56]. At given experimental conditions, the drug release rate is influenced by the intensity of interaction between the polymers chains and the NPs swelling behavior in a release medium as well as NPs size and shape. As shown in Figure 2C, EVE-2_4 and EVE-2_7 prolong the EVE release till 16 h or 48 h of incubation, respectively. EVE raw material dissolution in the same medium was completed at about 90 min, and in this time about 50% wt or the 37% wt of the loaded drug was released from NPs.

Furthermore, the quickest EVE release rate from EVE-2_4 sample (100% EVE released by 18th - 24th h), irrespective of the NPs larger sizes, may confirm the envisaged weaker polymers networks forming these NPs as also shown from TEM images (Figure 2A). Consistently, EVE-2_7 showed a prolonged release till the 48th h corroborating a more interconnected structure involving not only electrostatic interaction but also hydrogen bonds and van der Walls interactions. Hence, it can be speculated that stable HA/CS complexes also enables more efficient payload entrapment as previously highlighted (EVE-2_7 in Table 3).

For next biological studies, EVE-2_7 batch and the relative placebo Et_CS_7 batch were chosen.

### 3.3. Cellular uptake of HA/CS NPs

Carrier uptake mechanisms by tumor cells are important features affecting the therapeutic effect in cancer chemotherapy. One of the major challenges of nanomedicine is to build nanosystems able to deliver drugs directly and specifically to sick cells. In order to achieve this aim with maximum efficacy, nanocarriers interactions with cell and how they are internalized must be understood.

Fluorescent HA/CS-RhB NPs were used to follow the NPs trafficking over the cell membrane. With this purpose CS-RhB was synthetized, exploiting the RhB free carboxylic group and the high density of primary amine groups on CS backbone. The performed amidation reaction was characterized by a yield percentage of 44.2%.

UV–vis spectrum of CS-RhB conjugate was detected in a range of wavelengths of 200–700 nm: according to literature RhB molecule exhibits a high absorption peak at 554 nm [44,57] while no peak is observed for CS in the same range of wavenumbers. CS-RhB spectrum showed the characteristic peak at 554 nm confirming effective conjugation between CS and RhB took place (data not reported).

Next, the degree of labeling was calculated by measuring the absorbance of CS-RhB conjugate at 554 nm against RhB standard solutions in the 0.25–4 µg/mL concentration range. The calculated labeling efficiency was a mean of 0.267 µg of RhB/ µg of CS.

Lastly, a mixture of CS and CS-RhB conjugate (67% wt and 33% wt, respectively) was used to prepare fluorescent HA/CS-RhB NPs ensuring the formation of NPs with appropriate mean size of 199.92 ± 51.93 nm (PDI = 0.380 ± 0.033) and a ζ potential of −29.94 mV. The high negative surface charge highlights HA exposure on the NPs surface so the use of CS-RhB, in a suitable concentration, did not affect the polymers chains interaction during the microfluidic mixing.

Fluorescent CS-RhB/TPP NPs were prepared as control in order to discriminate the effect of HA coating on the uptake mechanism. CS-RhB/TPP NPs resulted in mean diameter of 245.35 ± 35.60 nm and PDI value below to 0.4 comparable with those of HA/CS-RhB NPs. As expected, CS-RhB/TPP NPs had a positive surface charge (+21.33 mV).

Figure 3A displays the cellular uptake of HA/CS-RhB NPs after 30 min of incubation with 50 and 100 µg/20,000 hMSCs. NPs are represented in the images as red dots; nuclei were stained by Hoechst (0.5 μg/mL) and they appear blue whereas CD44 glycoprotein on the cell surface was stained by using FITC labeled anti-CD44 antibody and it appears green. After 30 min of incubation HA/CS-RhB NPs were localized inside cell cytoplasm and near the perinuclear region: red stained NPs appear clearly surrounded by the green cell membrane.

Both HA/CS-RhB NPs concentrations (50–100 µg) (Figure 3A) showed more evident signal after 90 min incubation when fluorescent NPs are undoubtedly dispersed in the cytoplasmatic compartment. Furthermore, at 90 min incubation, endocytic vesicles are observed near plasma membrane. These entities were investigated more in details in the Figure 3B, representing the confocal optical sections of hMSCs containing 50 µg of NPs/20,000 cells HA/CS-RhB NPs at 90 min incubation. The histogram analysis of the fluorescence intensities along the yellow line across the selected vesicles (graph I, II in Figure 3B) can demonstrate that HA/CS-RhB NPs are confined to discrete cytoplasmic vesicle-like regions, approximately of 1–3 μm in diameter, placed near the CD44+ membrane surface. It can be speculated that HA/CS-RhB NPs interact with the cell membrane which, when invaginated, forms vesicles abled to transport the internalized material to a separate intracellular compartment. Vesicles formation is consistent, and it was also observed for the highest NPs concentration tested (100 µg/20,000 cells) after 90 min of incubation, as reported in Figure 3C.

Results concerning the HA/CS-RhB NPs internalization after 240 min of incubation are shown in Figure 3A. Confocal microscopy confirmed the notably high concentration of fluorescent NPs inside the cells and demonstrated that they are always localized in the cytoplasmic region never entering the nucleus. As highlighted before, after 4h of incubation HA/CS-RhB NPs were enclosed in cellular membrane vesicles useful for their internalization and located near the cell surface.

All confocal images were subsequently processed to determine the level of fluorescent NPs inside the cells and quantitatively evaluate the HA/CS-RhB NPs uptake in hMSCs. Quantitative results were expressed as CTCF and reported in Figure 3D.

It can be noted that fluorescence inside cells arises in the time frame from 30 min to 240 min. A remarkable increase in HA/CS-RhB NPs uptake was displayed after 4 h incubation regardless from the NPs concentration used and a statistical difference was highlighted by using Tukey’s multiple comparisons test (*p* value < 0.001).

HA/CS-RhB NPs internalization by hMSCs is undeniable even if no evident differences were observed by varying NPs concentration, and it is worth mentioning here that the present results demonstrate that HA/CS-RhB uptake was significantly improved with respect to HA/CS-RhB prepared by traditional method previously assessed [28]. The first remark that it can be provided is that the new microfluidic-based preparation process positively affects the HA/CS NPs biological performance.

Moreover, the present results allow to suggest the saturable endocytosis mechanism of the HA/CS-RhB NPs. This hypothesis is supported by other papers in literature [58,59,60,61] and indeed, it will be better addressed by further experiments.

The uptake profile of HA/CS-RhB NPs was compared to that of CS-RhB/TPP NPs to understand if HA, selective ligand for CD44, plays a pivotal role in the NPs cell internalization mechanism.

CS-RhB/TPP NPs appear to be predominantly localized in the intracellular compartment (Figure 3E) and the utmost internalization of CS-RhB/TPP NPs within cells was achieved by prolonging the incubation time to 240 min. 

The comparison between the quantitative uptake of HA/CS-RhB NPs and CS-RhB/TPP NPs are reported in Figure 3E. It can be noted that, the amount of the internalized HA/CS-RhB NPs seems to be higher for all considered time points. Sidak’s multiple comparisons test confirmed the statistical difference of CTCF between HA/CS-RhB NPs and CS-RhB/TPP NPs taken into hMSCs at 240 min of incubation. These results suggest the effective selectivity of HA for CD44 receptor maintained for on-chip synthetized HA/CS NPs and encourage us to deeply investigate these phenomena.

### 3.4. Competitive Binding Experiment

Now, it is well known that NPs with proper size can be internalized by several routes, in this case, considering our results, the most likely hypothesis is to be an endocytic pathway. To evaluate whether HA/CS NPs could have access to cells through the specific interaction with the HA transmembrane receptor, CD44, monoclonal antiCD44 anti-mouse antibody that selectively recognizes the N-terminal domain of CD44 and inhibits the HA binding, was used. As observed in Figure 4, preventing HA/CD44 interaction the red fluorescence of HA/CS-RhB NPs inside the hMSC was significantly reduced (*p* value = 0.008). The HA/CS-RhB NPs uptake was almost negligible after the receptor blockage and it may be assumed that CD44 is responsible for the receptor mediated uptake.

On another hand, blocking the CD44, CS-RhB/TPP NPs were massively internalized in the cell cytosol, quantitative data being comparable to those shown without antiCD44 antibody.

These results suggest that CS-RhB/TPP NPs uptake happens regardless of CD44 receptor.

Consequently, we concluded that the internalization of on-chip synthetized HA/CS NPs may occur by CD44 receptor-mediated endocytosis.

### 3.5. In Vitro Cytotoxicity

Figure 5 shows hMSCs cytotoxicity data under three different experimental conditions including incubation with increasing concentration of EVE loaded HA/CS NPs, incubation with equal dose of free EVE (EVE 0.01 to 88.12 µg/mL) and placebo HA/CS NPs (0.1–1000 µg/mL). After culture for 24 h, fewer cells were detected in free drug groups, compared with the same dose of drug-loaded NPs groups. At the highest concentration of EVE (88.12 µg/mL) the cell viability percentage was 14.86 ± 2.80% for EVE loaded NPs meanwhile no viable cells were observed for EVE solution. Therefore, it may be claimed that EVE loaded HA/CS NPs were less toxic respect to free EVE, well known for its narrow therapeutic index and liable of severe side effects. Moreover, it was observed that the cytotoxicity effect was dose dependent; for both free EVE and EVE loaded NPs cell viability was dramatically reduced increasing EVE concentration. From the fitted curve, IC50 of the free drug, 3.476 µg/mL with *R*^2^ of 97%, resulted to be 5 times more toxic than EVE loaded HA/CS NPs (15.99 µg/mL with *R*^2^ of 98%). The reduction of cell viability is accompanied by their morphological change. Morphological examination revealed cells in severe distress. hMSCs displayed a flattened and round shape and they failed to reach the confluence state.

The results of in vitro assay provide an evidence that placebo HA/CS NPs were not cytotoxic at the concentrations tested, viability percentage was always higher than 70% confirming the well-known biocompatibility of HA and CS. The quantitative results were supported by morphological analysis, placebo HA/CS NPs appeared to be safe, no toxic effects arose, and after incubation, cells maintained the typical confluence state.

### 3.6. Assessment of DNA Synthesis and Proliferative Activity by 5-Bromo-2′-Deoxyuridine (BrdU) Incorporation

The ability of EVE loaded HA/CS NPs to inhibit the hMSCs proliferation was determined in vitro by BrdU assay. BrdU can be incorporated into new synthetized DNA during the S phase of cellular cycle. Using a proper primary monoclonal antibody anti-bromodeoxyuridine and its secondary anti-mouse antibody marked with FITC, proliferating cells can be observed and counted. hMSCs were treated with different concentrations of EVE loaded HA/CS NPs (from 5 to 50 μg/20,000 cells) and the treatment was followed over time (30, 90 and 240 min). Proliferating activity was evaluated and expressed as % BrdU cells positive. Placebo HA/CS NPs (10–50 μg/20,000 cells) and EVE solution (EVE sol, 4.4 μg/mL) were used as positive controls while untreated hMSCs (CTR) were chosen as negative control.

Figure 6A shows the results of hMSCs proliferation after 30 min of treatment. Data show that 50 µg/20,000 cells of EVE loaded HA/CS NPs prominently inhibit hMSCs proliferation compared to the same concentration of placebo HA/CS NPs. As highlighted in Figure 6A, statistical analysis shows a significant proliferation difference expressed by the *p* value of 0.039. Statistically significant proliferation differences were also found among the cells treated with increasing concentrations of EVE loaded HA/CS NPs. Comparison by multiple T-tests showed that cells proliferation inhibition was substantially stronger using the highest EVE loaded HA/CS NPs concentrations (25–50 µg/20,000 cells); *p* values resulted to be 0.017 and 0.012, respectively, when compared to proliferation inhibition provided by 5 µg of NPs/20,000 cells. Meanwhile, using EVE solution at 4.4 µg/mL no proliferating cells were observed.

By assessing results reported in Figure 6B for 90 min of incubation, statistical difference in cells proliferation was highlighted after treatment with placebo HA/CS NPs and EVE loaded HA/CS NPs at different concentrations, namely 25 µg/20,000 cells and 50 µg/20,000 cells. Multiple *t*-test revealed *p* values of 0.008 and 0.002, respectively. In 90 min timing, it is interesting to note that EVE solution and EVE loaded HA/CS NPs (10–25–50 µg/20,000 cells) have overlapping effectiveness on cells proliferation inhibition.

Lastly, hMSCs proliferation results, after 240 min of incubation, are shown in Figure 6C.

First, it is evident a significant gap in cells proliferation between EVE solution and EVE loaded HA/CS NPs (25 µg and 50 µg/20,000 cells). After 240 min incubation, EVE loaded HA/CS NPs achieved a more effective inhibition of hMSCs proliferation if compared to the free drug and no proliferative cells were observed for both EVE loaded NPs concentrations. Comparing data obtained for EVE solution and EVE loaded HA/CS NPs (25 µg and 50 µg/20,000 cells), *p* value of 0.037 and 0.012 were determined, respectively. As stated before, placebo HA/CS NPs and EVE loaded HA/CS. NPs at the same concentration (50 µg/20,000 cells) caused a remarkable difference (*p* value = 0.016) in cells proliferation.

## 4. Conclusions

In this study, we demonstrated that SHM-assisted ionotropic gelation method is a robust, reproducible technique to precisely induce electrostatic interactions between HA and CS. Well-controlled on-chip HA/CS NPs synthesis ensured to produce NPs with tailored size distribution for various administration routes, effectively loading EVE, a challenging drug because of its poor water solubility, and controlling its release.

Additionally, results support the concept that on-chip synthetized EVE loaded HA/CS NPs effectively crossed the cell membrane via CD44 receptor, releasing EVE in the cell cytosol and notably reducing the proliferative activity of CD44+ expressing cells. More in details, regarding the NPs internalization mechanism HA coating resulted to be important. Both uncoated (CS/TPP NPs) and HA-based NPs (HA/CS NPs) were internalized through an active mechanism at least partially based on endocytic vesicles formation, however, we demonstrated the different internalization manner that is ascribed to the different binding form to the cell membrane. CS/TPP NPs exploit a non-specific electrostatic attraction whereas HA/CS NPs use a more specific binding to CD44 receptor.

The microfluidic method allows preparing formulations under a less hardworking process using small amounts of raw materials and with reduced product waste. Moreover, it permits to obtain NPs batches in a high-throughput manner by parallelizing individual SHM units, while maintaining the advantages of the set-up on-chip formulation procedure without altering the final NPs characteristics. SHM-assisted ionotropic gelation method reveals a versatile technique for HA/CS NPs one-step synthesis and a promising candidate for replacing the conventional bulk mixing method techniques, hopefully contributing to a straightforward translation of HA-based NPs from bench to clinic.

## Figures and Tables

**Figure 1 pharmaceutics-12-00260-f001:**
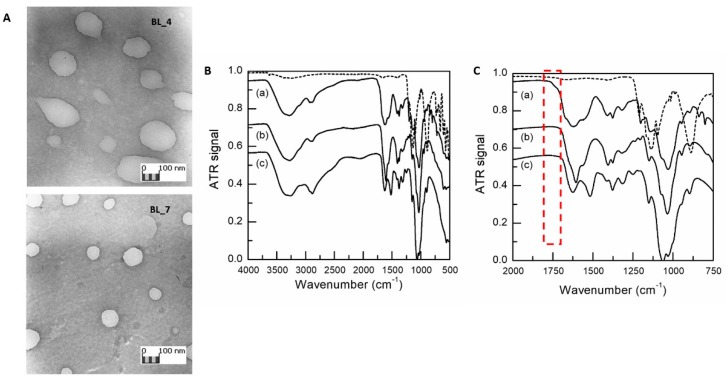
(**A**) TEM images of BL_4 and BL_7 samples (obj. mag. 20 k×). (**B**) FTIR spectra from 500 to 4000 cm^−1^ of sodium tripolyphosphate (TPP) raw material (dashed line), freeze dried HA/CS NPs (Et_CS_7) (a), HA raw material (b) and CS raw material (c); (**C**) Enlargement of the fingerprint region from 750 to 2000 cm^−1^. HA and CS spectrum have been shifted vertically by -0.2 and -0.45, respectively, to clarify viewing.

**Figure 2 pharmaceutics-12-00260-f002:**
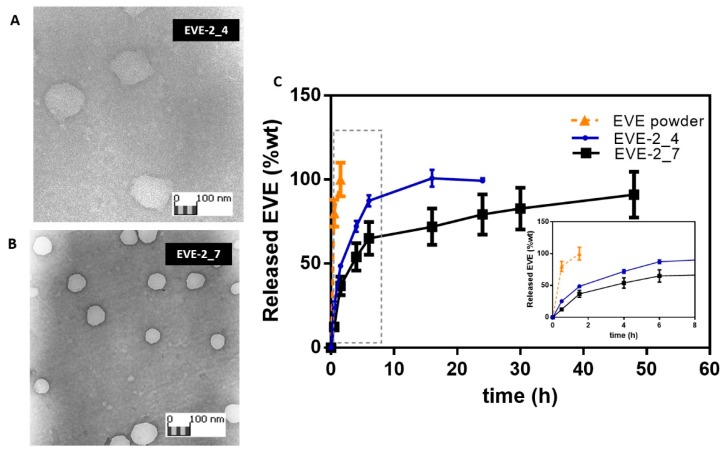
TEM images of EVE loaded HA/CS NPs, EVE-2_4 (**A**) and EVE-2_7 (**B**). (**C**) In vitro EVE release from EVE-2_4 (blue line) and EVE-2_7 (black line), 100 μg of NPs was suspended in 0.01M PBS with 1% *w*/*v* Tween 20 pH 7.4. NPs release behavior is compared to EVE raw material dissolution profile (orange line). Insert: EVE release profiles between 0 and 8 h of incubation.

**Figure 3 pharmaceutics-12-00260-f003:**
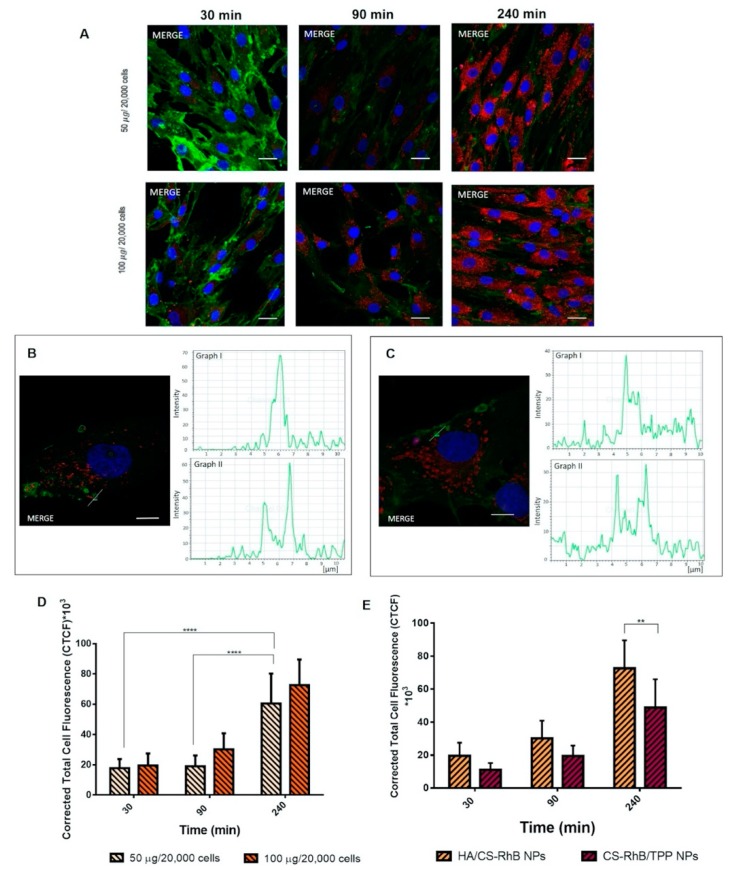
(**A**) Confocal microscopy images of human mesenchymal stem cells (hMSCs) after 30, 90 and 240 min of incubation at 37 °C with different concentration of HA/CS-RhB NPs (50–100 μg/20,000 cells). Red fluorescent denotes HA/CS NPs; nuclear blue fluorescence of DNA with Hoechst33258 dye; positive expression for cluster of differentiation-44 (CD44) on hMSCs membrane was detected by using anti-CD44 primary antibody and FITC labeled secondary antibody (green fluorescence), scale bar = 20 µm. (**B**) Detail of the endocytic vesicles formation: confocal image of hMSCs after 90 min incubation with HA/CS-RhB NPs (50 µg/20,000 hMSCs), a yellow line, crossing a vesicle, was drawn and the analysis of the red and green fluorescence intensities across this line was reported in graph I and II, respectively (scale bar = 10 µm). (**C**) Detail of the endocytic vesicles formation: confocal image of hMSCs after 90 min incubation with HA/CS-RhB NPs (100 µg/20,000 hMSCs), a yellow line, crossing a vesicle, was drawn and the analysis of the red and green fluorescence intensities across this line was reported in graph I and II, respectively (scale bar = 10 µm). (**D**) Effect of treatment time on red fluorescent intensity of the internalized HA/CS-RhB NPs (50–100 μg/20,000 cells). Results are presented as mean ± SD.**** *p*< 0.001. (**E**) Red fluorescent intensities of HA/CS-RhB NPs and CS-RhB/TPP NPs (100 μg/20,000 hMSCs) localized in the cell cytoplasm after 30, 90, 240 min of incubation. Results are presented as mean ± SD. ** *p* value < 0.01.

**Figure 4 pharmaceutics-12-00260-f004:**
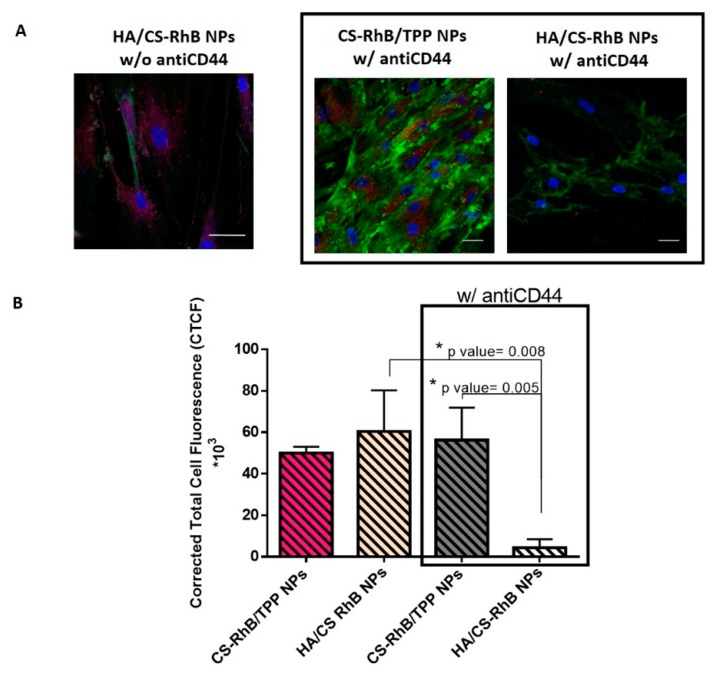
(**A**) Confocal microscopy images of hMSCs incubated with HA/CS-RhB NPs, CS-RhB/TPP NPs (50 μg/20,000 hMSCs) for 240 min with or without monoclonal antiCD44 primary antibody (scale bar = 20 μm). (**B**) The red fluorescent intensity of HA/CS-RhB NPs and CS-RhB/TPP NPs localized in the cell cytosol. Results are presented as mean ± SD. Holm-Sidak multicomparison method, * *p* value < 0.05.

**Figure 5 pharmaceutics-12-00260-f005:**
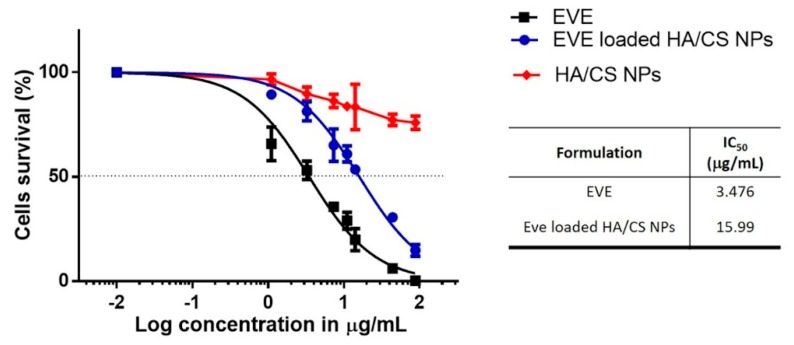
Cytotoxicity results obtained by incubating hMSCs with EVE loaded HA/CS NPs, HA/CS NPs and free EVE. The graph shows the cell viability curves for the experimental conditions fitted with a sigmoidal dose-response curve. The resulting IC50 values are shown in the table. Results were expressed as mean ± SD.

**Figure 6 pharmaceutics-12-00260-f006:**
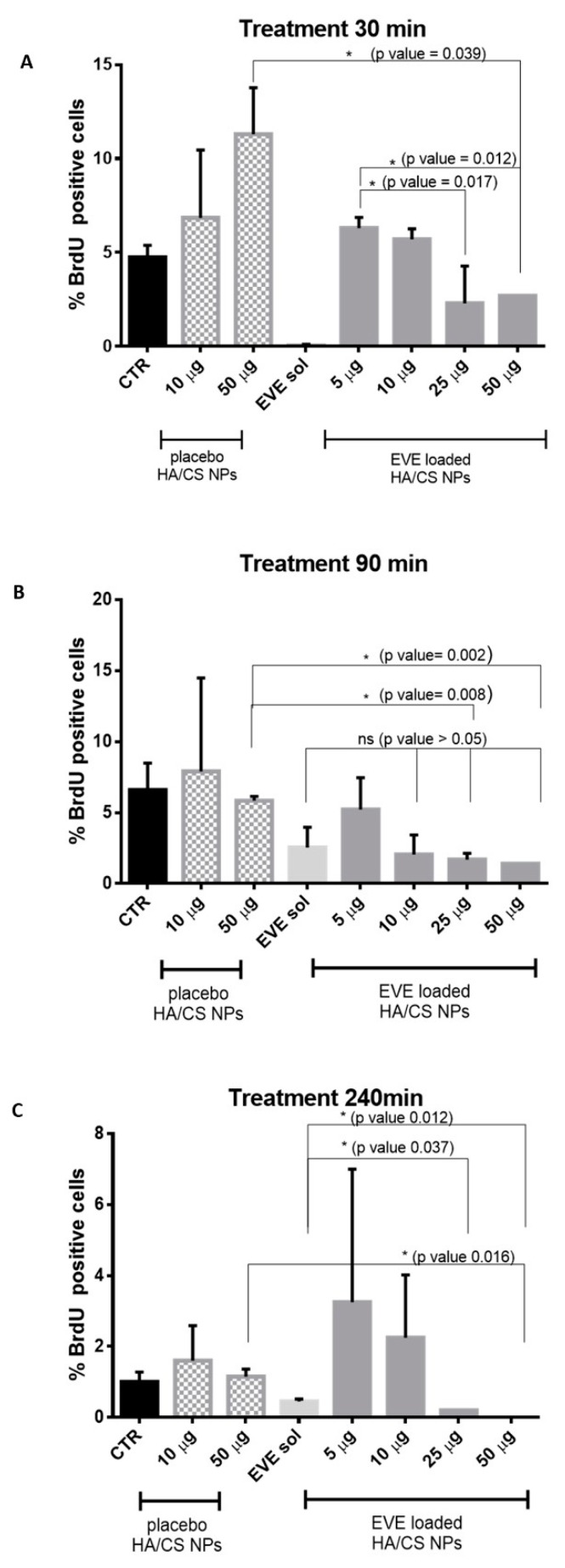
Effect on cell proliferation of placebo HA/CS NPs (10 and 50 µg/20,000 cells), EVE solution (EVE sol, 2.2 μg/20,000 cells) and EVE loaded HA/CS NPs after 30 min (**A**), 90 min (**B**) and 240 min (**C**) of treatment.

**Table 1 pharmaceutics-12-00260-t001:** Set-up of placebo hyaluronic acid (HA)/chitosan (CS) nanoparticles (NPs) on-chip preparation method: mean size (nm), polydispersity index (PDI), ζ potential (mV).

Sample Code	HA Conc (mg/mL)	HA:CS (*w*:*w*)	CS:TPP (*w*:*w*)	Mean Size ± SD (nm)	PDI ±SD	ζ Potential ± SD (mV)
BL_1	0.400	4:1	25:1	1025.45 ± 255.10	0.486 ± 0.045	−24.88 ± 3.07
BL_2	0.150	1.5:1	25:1	102.75 ± 3.23	0.297 ± 0.007	+9.30 ± 1.89
BL_3	0.150	1.5:1	50:1	159.85 ± 38.91	0.313 ± 0.039	+17.34 ±4.13
BL_4	0.400	8:1	12.5:1	584.00 ± 22.20	0.422 ± 0.027	−24.02 ± 2.91
BL_5	0.400	8:1	25:1	906.55± 103.21	0.393 ± 0.046	−18.01 ± 8.31
BL_6	0.150	3:1	12.5:1	239.20 ± 19.51	0.520 ± 0.059	−21.79 ± 10.64
BL_7	0.150	3:1	25:1	159.00 ± 7.28	0.306 ± 0.053	−20.85 ± 3.16
BL_8	0.275	3.6:1	25:1	758.45 ± 23.77	0.457 ± 0.055	−23.89 ± 3.34

**Table 2 pharmaceutics-12-00260-t002:** Comparison of mean size, PDI and ζ potential and polymeric composition of different placebo HA/CS NPs batches prepared by adding Et or Me in CS solution.

Sample Code	Mean Size ± SD (nm)	PDI ± SD	ζ Potential ± SD (mV)	CS Amount ± SD (μg)	HA Amount ± SD (μg)
Et_CS_4	349.15±38.09	0.473 ± 0.058	−25.80 ± 3.81	11.03 ± 3.23	238.21 ± 42.45
Me_CS_4	406.80 ± 19.65	0.467 ± 0.012	−31.72 ± 0.52	25.96 ± 17.92	189.42 ± 72.88
Et_CS_7	117.50 ± 4.51	0.265 ± 0.024	−22.52 ± 3.43	17.56 ± 3.96	220.76 ± 25.74
Me_CS_7	120.03 ± 13.92	0.315 ± 0.025	−20.18 ± 2.38	14.76 ± 6.46	222.69 ± 22.35

**Table 3 pharmaceutics-12-00260-t003:** Physical characterization and Everolimus (EVE) encapsulation of EVE loaded HA/CS NPs prepared using different amount of EVE (*n* = 6).

Sample Code	CS/EVE Ratio	Mean Size ± SD (nm)	PDI ± SD	ζ Potential ± SD (mV)	DL ± SD (μg of EVE / 1 mg of NPs)	EE% ± SD
EVE-1_4	2:1	372.35 ± 71.50	0.440 ± 0.041	−24.61 ± 7.26	33.09 ± 3.66	35.29 ± 3.90
EVE-2_4	5:1	415.58 ± 84.56	0.490 ± 0.061	−28.71 ± 2.78	48.91 ± 26.33	56.05 ± 11.91
EVE-1_7	2:1	144.93 ± 13.31	0.286 ± 0.059	−28.91 ± 6.18	78.81 ± 6.42	29.32 ± 10.64
EVE-2_7	5:1	135.07 ± 45.81	0.356 ± 0.091	−17.28 ± 2.30	88.12 ± 20.76	54.56 ± 7.45

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
