# Peer review of "On-Chip Synthesis of Hyaluronic Acid-Based Nanoparticles for Selective Inhibition of CD44+ Human Mesenchymal Stem Cell Proliferation"

_pharmaceutics, 2020, doi:10.3390/pharmaceutics12030260_

Round 1

Reviewer 1 Report

In this study, authors developed a microfluidics based method for one-step synthesis of hyaluronic acid based nanoparticles. The preparation is novel, and would be useful for other researchers. Therefore, it can be accepted after the comments were addressed. 1. In the Introduction, authors should clearly describe the advantages of CS in constructing hybrid nanoparticles. 2. Although the method could achieve very high drug loading capacity, the encapsulation efficiency was not high ,even at lower DLX. Authors should discuss it. 3. In table 3, Higher CS/EVE means lower concentration of EVE, but why was the DL of EVE-2 higher than that of EVE-1. It is unbelievable. 4. In figure 4, why did authors not provide uptake of CS-RhB/TPP NPs without antiCD44? 5. In Figure 3e, the uptake of CS-RhB/TPP NPs was significantly lower than HA/CS-RhB NPs. But in Figure 4,HA/CS-RhB NPs was comparable to CS-RhB/TPP NPs. 6. There are several papers used HA for drug delivery (Acta Pharm Sin B, 2019, 9(2):410-420; Adv Funct Mater 2018;28:1804490; Nanomedicine 2016;11:2341-57.), authors should refer them in Introduction.

Author Response

Authors thank the reviewer for her/his careful reading of the manuscript and constructive remarks.

Please find below a detailed point-by-point response (in blue) to all comments (in black).

Reviewer 1

In this study, authors developed a microfluidics based method for one-step synthesis of hyaluronic acid based nanoparticles. The preparation is novel, and would be useful for other researchers. Therefore, it can be accepted after the comments were addressed.

1. In the Introduction, authors should clearly describe the advantages of CS in constructing hybrid nanoparticles.

According to the reviewer request, the advantages of CS, from regulatory/pharmaceutical and biological standpoint, are better explained in the Introduction (Line 77-84) and new references were included.

Line 77-84: “Chitosan (CS) is a natural polysaccharide obtained from renewable animal or vegetable sources, and it has been the most extensively used cationic polymer combined to HA in HA-based NPs (HA/CS NPs) preparation [20-24]. Thanks to its well-known non-toxicity, biocompatibility, biodegradability and low immunogenicity, CS recently gained FDA-approval for human use. CS suitability as nanomedicine ingredient is further broadened by its biological properties like mucoadhesion, antimicrobial, antifungal and antioxidant activities, but above all its inherent capacity to transiently open tight junctions [20,25].”

2. Although the method could achieve very high drug loading capacity, the encapsulation efficiency was not high ,even at lower DLX. Authors should discuss it. 3. In table 3, Higher CS/EVE means lower concentration of EVE, but why was the DL of EVE-2 higher than that of EVE-1. It is unbelievable.

Authors agree with the referee that EE% seems not so high in absolute terms, it is however significantly improved if compared to the encapsulation efficiency obtained for EVE loaded HA/CS NPs prepared by traditional bulk mixing method (Chiesa E. et al., Int. J. Mol. Sci. 2018, 19, 2310 – ref. 28 in the revised manuscript). This evidence is better underlined in the revised manuscript (Line 636-638)

Considering NPs with different polymer composition, for the highest CS/EVE (5:1) tested the highest EE% values were obtained (>55%).

We can speculate that a further increase of the CS/EVE ratio (more than 5:1) may lead to a higher EE%, as observed in other works (Forbes, N. et al. Int. J. Pharm., 2019, 556, 68-81 – ref. 55 in the revised manuscript).

EVE loading was similar in HA/CS NPs with similar composition in terms of HA and CS independently from the CS/EVE (2:1 – 5:1 wt) and the highest DLs were achieved for NPs prepared containing a higher amount of CS. This behaviour can be attributed to nucleation process and polyelectrolytes interactions entity between HA and CS during NPs formation inside microfluidic device (Russo, M. et al. Sci. Rep. 2016, 6, 37906; Geven, M. et al., Material Matters™, 2019, 14, 83-89; Borro, B.C. et al., J. Colloid. Interf. Sci. 2019, 538, 559–568 - refs. 35,53,54 in the revised manuscript) (Line 624-635).

The overall discussion on EVE loading and encapsulation efficiency issues has been more thoroughly addressed in the revised manuscript (we apologize for a typing error in Table 3, now amended) (Line 624-638).

Line 624-638: “Considering the challenging features of the drug and the hydrophilic polymeric network, satisfactory EVE loading was achieved for all batches: drug loading was increased from 30- to 90-fold if compared to EVE loading obtained in HA/CS NPs prepared by a traditional bulk mixing method [28].

DLs of NPs with the same HA and CS composition, namely EVE-1_4 - EVE-2_4 and EVE-1_7-EVE-2_7 (Table 3), were not affected by the two CS/EVE ratios tested, however the results demonstrate the relevant role of CS amount to obtain the highest EVE loadings. In particular, for the same CS/EVE ratio (EVE-1_7 vs EVE-1_4 and EVE-2_7 vs EVE-2_4) the NPs characterized by a greater CS content (EVE-1_7 and EVE-2_7) reached the highest DLs: in presence of a non-ionic small drug this behaviour can be ascribed to the role of CS molecules in the polyelectrolyte complexation of HA inside the microfluidic channel during the nucleation process typically depending on both anionic/cationic macromolecules concentration and size as well as their balance [35,53,54].

Nevertheless, for NPs with different polymer composition, a lower CS/EVE (2:1) ratio (EVE-1_4, EVE-1_7) induced a drop in the EE%, as previously reported in literature [55]. EVE-2_4 and EVE-2_7 reached the most satisfactory EE% (> 55%).”

4. In figure 4, why did authors not provide uptake of CS-RhB/TPP NPs without antiCD44?

According to the referee request, image of uptake of CS-RhB/TPP NPs without antiCD44 was properly added in Figure 4.

5. In Figure 3e, the uptake of CS-RhB/TPP NPs was significantly lower than HA/CS-RhB NPs. But in Figure 4,HA/CS-RhB NPs was comparable to CS-RhB/TPP NPs.

The authors thank the referee to point out this peculiar behaviour and they correlated this difference to the different experimental procedures. In the first case (Figure 3E), hMSCs were incubated at 37°C and then treated with NPs; in the other case (Figure 4) hMSCs were incubated at 4°C for 30 min, with or without antiCD44 antibody, and then they were allowed to warm to 37°C and treated with NPs. Incubation at 4°C (also without antiCD44 antibody) could influence the HA/CS-RhB NPs uptake and this behaviour will be pointed out in a subsequent and deeper uptake study concerning the several mechanisms acting in this uptake process. The main goal of the tests developed is to highlight the lead role of CD44 in selective internalization of HA/CS NPs synthetized by microfluidic-based ionotropic gelation.

Furthemore, the NPs concentration was not comparable: "100 µg/20,000 cells in Figure 3E and 50 µg/20,000 cells in Figure 4; to better clarify, NPs concentrations are now indicated in both figures’ captions (Line 784 and 804-805).

6. There are several papers used HA for drug delivery (Acta Pharm Sin B, 2019, 9(2):410-420; Adv Funct Mater 2018;28:1804490; Nanomedicine 2016;11:2341-57.), authors should refer them in Introduction.

The suggested references were added in Introduction of the revised manuscript.

Reviewer 2 Report

In this study, the authors have developed an innovative microfluidics-based method for one-step synthesis of hyaluronic acid (HA)-based nanoparticles. They observed that HA/CS NPs loaded with EVE effectively crossed the cell membrane via CD44 receptor, released EVE in the cell cytosol and reduced the proliferative activity of CD44+ expressing cells. The studies partially support this proposal, but some methodological and technical issues limit the reliability of the conclusions. There are several points that need other studies and clarification before this manuscript could be recommended for publication. The major revisions are needed in following points:

1.) The manuscript needs editing and rewrites throughout the document. The authors must seek professional editing assistance from native English speakers or professionals.

2.) One of the major point of criticism concern the study of DNA synthesis and proliferative activity by BrdU incorporation. It should be noted that DNA synthesis analysis makes sense in proliferating cells. The authors stated that they maintained cells until they reached confluence (line 343) and subsequently treated cells with NPs. In such condition, the cells begin to be silenced and stop proliferating. Moreover, I understand that the division of hMSCs cells takes close to 24 hours. If so, I do not understand why the authors chose such short time intervals, i.e. 30-240 min.

3.) My additional concern is how the authors performed cellular uptake of NPs. Why the Authors did not decide to use lower NPs concentration, especially because they were aware of the fact that the tested mechanisms may be saturable. I suggest to perform additional cellular uptake experiments with lower NPs concentration.

4.) In line 373: Does the one-way term refer to Anova test?

5.) In Figure 3, it is absolutely necessary to show a better quality and representative images. Now it is difficult to read especially images B and C. Additionally, scale bar should be showed.

6.) In Figure 4A scale bar is missing.

7.) In Figure 5. It is impossible to calculate IC50 for HA/CS NPs as long as survival curve did not reach the 50% of survival.

Round 2

Reviewer 2 Report

I appreciate the authors put much effort to fix the issues that were mentioned in the original review. The corrections implemented by the Authors improve the quality of the manuscript.